# Pelvic Lymphadenectomy May Not Improve Biochemical Recurrence-Free Survival in Patients with Prostate Cancer Treated with Robot-Assisted Radical Prostatectomy in Japan (The MSUG94 Group)

**DOI:** 10.3390/cancers14235803

**Published:** 2022-11-25

**Authors:** Sanae Namiki, Makoto Kawase, Shin Ebara, Tomoyuki Tatenuma, Takeshi Sasaki, Yoshinori Ikehata, Akinori Nakayama, Masahiro Toide, Tatsuaki Yoneda, Kazushige Sakaguchi, Jun Teishima, Kazuhide Makiyama, Takahiro Inoue, Hiroshi Kitamura, Kazutaka Saito, Fumitaka Koga, Shinji Urakami, Takuya Koie

**Affiliations:** 1Department of Urology, Gifu University Graduate School of Medicine, Gifu 5011194, Japan; 2Department of Urology, Hiroshima City Hiroshima Citizens Hospital, Hiroshima 7308518, Japan; 3Department of Urology, Yokohama City University, Yokohama 2360004, Japan; 4Department of Nephro-Urologic Surgery and Andrology, Mie University Graduate School of Medicine, Tsu 5148507, Japan; 5Department of Urology, University of Toyama, Toyama 9300194, Japan; 6Department of Urology, Dokkyo Medical University Saitama Medical Center, Koshigaya 3438555, Japan; 7Department of Urology, Tokyo Metropolitan Cancer and Infectious Diseases Center Komagome Hospital, Tokyo 1138677, Japan; 8Department of Urology, Seirei Hamamatsu General Hospital, Hamamatsu 4308558, Japan; 9Department of Urology, Toranomon Hospital, Tokyo 1058470, Japan; 10Department of Urology, Kobe City Hospital Organization Kobe City Medical Center West Hospital, Kobe 6530013, Japan

**Keywords:** multicenter cohort study, pelvic lymph node dissection, prostate cancer, robot-assisted radical prostatectomy

## Abstract

**Simple Summary:**

This study evaluated the utility of pelvic lymph node dissection (PLND) in prostate cancer (PCa) patients undergoing robot-assisted radical prostatectomy (RARP). After propensity score matching, 1210 patients were enrolled and divided into two groups: those who underwent RARP without PLND (non-PLND group) and those who underwent PLND (PLND group). At the end of the follow-up period, no deaths due to PCa were identified in this study. Seventy-one patients (5.9%) had biochemical recurrence after RARP, and the 2-year biochemical recurrence-free survival (BRFS) rate was 95.0% for all patients, 95.8% for the non-PLND group and 94.3% for the PLND group (*p* = 0.855). For the all-risk group, there were no significant differences between patients who did and did not undergo PLND. Nevertheless, the results of the log-rank study indicate that PLND may be unnecessary for patients with PCa undergoing RARP.

**Abstract:**

In this multicenter retrospective cohort study, we aimed to evaluate whether pelvic lymph node dissection (PLND) improved biochemical recurrence (BCR) in patients with prostate cancer (PCa) who underwent robot-assisted radical prostatectomy (RARP) in Japan. A multicenter retrospective cohort study of 3195 PCa patients undergoing RARP at nine institutions in Japan was conducted. Enrolled patients were divided into two groups: those who underwent RARP without PLND (non-PLND group) and those who underwent PLND (PLND group). The primary endpoint was biochemical recurrence-free survival (BRFS) in PCa patients who underwent PLND. We developed a propensity score analysis to reduce the effects of selection bias and potential confounding factors. Propensity score matching resulted in 1210 patients being enrolled in the study. The 2-year BRFS rate was 95.0% for all patients, 95.8% for the non-PLND group, and 94.3% for the PLND group (*p* = 0.855). For the all-risk group according to the National Comprehensive Cancer Network risk stratification, there were no significant differences between patients who did and did not undergo PLND. Based on the results of the log-rank study, PLND may be unnecessary for patients with PCa undergoing RARP.

## 1. Introduction

Robot-assisted radical prostatectomy (RARP) is one of the minimally invasive surgical options for localized or selected advanced prostate cancers (PCa) based on various guidelines [1,2,3]. Pelvic lymph node dissection (PLND) is also an integral component of radical prostatectomy (RP) in patients with intermediate- or high-risk PCa [2,4]. PLND plays an important role in the accurate diagnosis of PCa staging and lymph node involvement (LNI) and may have potential benefits for eradicating micrometastases or regional LNI [5,6]. According to the European Association of Urology (EAU) guidelines, patients with intermediate-risk PCa with an LNI probability of 5% or greater, based on the Briganti nomogram [7], or those with high-risk PCa should receive extended PLND (ePLND) [1]. Likewise, the National Comprehensive Cancer Network (NCCN) guidelines recommend ePLND for patients with intermediate- and high-risk PCa [3]. However, the American Urological Association guidelines acknowledge that although PLND is the most effective procedure to detect LNI, there is no evidence of its therapeutic benefits [3]. Using Briganti nomogram cutoffs of 5% and 7%, 195 (46.4%) and 235 (55.9%) patients could have avoided ePLND, while 4 (2.1%) and 6 (2.6%) could have missed LNI [8]. Therefore, the therapeutic efficacy of PLND remains controversial [6].

To date, the effect of PLND on the oncological outcomes, biochemical recurrence (BCR), metastasis, and cancer-specific mortality in PCa remains unclear [4,5,6,9,10]. Several recent studies have demonstrated that the presence/absence or extent of PLND has no statistical impact on oncological outcomes such as biochemical recurrence-free survival (BRFS) [4,9,10]. Furthermore, ePLND was significantly correlated with increased perioperative and postoperative complications such as blood loss, lymphocele, thromboembolic events, and longer operative time [11,12,13]. In fact, PLND is reported to account for 10% of the operative time, and it is clear that the operative time is longer with PLND than without [13]. Colicchia et al. reported that ePLND was found to increase the operative time required for PLND (47 versus 26 min) compared to limited or standard PLND [14]. Conversely, several studies have shown that ePLND may be associated with improved cancer control or overall survival (OS) in patients with PCa at a high risk of LNI [15,16].

The purpose of this multicenter retrospective cohort study was to evaluate whether PLND improves BCR in PCa patients undergoing RARP in Japan.

## 2. Materials and Methods

### 2.1. Patient Population

The Institutional Review Board of Gifu University (approval number: 2021-A050) has approved this study. Since this study is being conducted retrospectively, it is not necessary to obtain informed consent from the patients. Since the results of retrospective and observational studies using existing materials and other data have already been published, written consent was not required in accordance with the provisions of the Japanese Ethics Committee and Ethical Guidelines. The details of the study can be found at https://www.med.gifu-u.ac.jp/visitors/disclosure/docs/2021-B039.pdf (accessed on 30 September 2022).

A retrospective multicenter cohort study of 3195 PCa patients (MSUG94 cohort) who underwent RARP at 9 domestic centers from September 2012 to August 2021 was conducted. The following clinical data were collected on enrolled patients: age, height, weight, serum initial prostate-specific antigen (PSA) level, biopsy Gleason grade (GG) on biopsy, clinical stage, risk stratification using NCCN criteria [2], the Eastern Cooperative Oncology Group performance status(ECOG-PS), and whether neoadjuvant or adjuvant therapy is administered The following surgical outcomes and pathological characteristics were recorded: console time, estimated blood loss, GG of surgical specimens, tumor (T) and node (N) stages of the surgical specimens, number of dissected lymph nodes, and Presence of positive resection margins. Tumors were staged according to the American Joint Committee on Cancer 8th Edition “Cancer Staging Manual” [17]. No data were collected in this study on whether enrolled patients underwent magnetic resonance imaging evaluation prior to prostate biopsy. Based on the International Society of Urologic Pathology (ISUP) 2014 guidelines [18], biopsy GGs were evaluated and classified into five groups. 

In this study, RARP was performed on all enrolled patients. The presence and extent of PLND, and nerve preservation approaches were determined at the discretion of the surgeon or according to each institution’s policy; the extent of PLND was categorized as limited (only obturator LN) or extended (performed to the common iliac vascular ureteric crossing, including or excluding the presacral lymph nodes) [19,20]. Enrolled participants were divided into two groups: those who underwent RARP without PLND (non-PLND group) and those who underwent PLND (PLND group).

### 2.2. Pathological Analysis

All prostatectomy specimens were evaluated by whole mount staining according to ISUP 2005 guidelines [21]. The apex of the prostate was cut perpendicular to the prostatic urethra for pathological evaluation. The end of the bladder neck was cut out of the excised specimen in a conical shape and cut perpendicular to it to prepare the pathology specimen. The remaining prostate tissue was sectioned perpendicular to the urethra at 3–5 mm intervals and evaluated pathologically.

### 2.3. Follow-Up Schedule

All patients were evaluated for serum PSA and testosterone levels at 3-month postoperative intervals. Patients with postoperative serum PSA levels rising to 0.2 ng/mL were classified as having a BCR. If the postoperative PSA level was not less than 0.2 ng/mL, the date of RARP was considered as the time of BCR.

### 2.4. Endpoints and Statistical Analyses

The primary endpoint was BRFS in PCa patients who underwent PLND. BCRs in the presence or absence of PLND, as classified by the NCCN risk stratification, were set as the secondary endpoints. Data were analyzed using SPSS software (version 24.0; IBM Corp., Armonk, NY, USA). Continuous variables were compared using Student’s *t*-test, and categorical variables were compared using Fisher’s exact test or the McNemar test.

Comparing the PLND and non-PLND groups, the PLND group had significantly higher initial PSA, biopsy GG (*p* < 0.0001), and significantly more patients with clinical T3 and high-risk PCa (*p* < 0.0001). For these reasons, we considered it likely that differences in patient background between the two groups would be reflected in the results of this study. Therefore, we developed a propensity score analysis to reduce the effects of selection bias and potential confounding factors in this study. A propensity score was calculated for each patient using multivariate logistic regression, with age, biopsy and pathologic GG, clinical stage, NCCN risk stratification, and PSM set as covariates. This method accounts for the confounding imbalances between individual study cohorts. Propensity scores were calculated for all participants using a combination of continuous and categorical factors. Individuals from each study cohort were then matched with individuals from the reference cohort according to the calculated propensity scores. BRFS after RARP was evaluated using the Kaplan–Meier method. The relationship between BCR and subgroup classification was analyzed using the log-rank test. All *p*-values were two-sided, and *p*-values < 0.05 were considered statistically significant.

## 3. Results

### 3.1. Patient Characteristics

A total of 1210 patients were enrolled in the study. The median age was 69 years (interquartile range [IQR], 64–73 years), body mass index was 23.4 kg/m^2^ (IQR, 21.7–25.4 kg/m^2^), and PSA was 7.100 ng/mL (IQR, 5.300–10.300 ng/mL). Biopsy GG ≥ 3 and clinical T3 were performed in 336 (27.8%) and 28 (2.3%) patients, respectively. According to the NCCN risk stratification, low-, intermediate-, and high-risk PCa risk were found in 181 (15.0%), 852 (70.3%), and 178 (14.7%) patients, respectively. Thirty-four patients (2.8%) had an ECOG-PS score of ≥1. The median follow-up period was 22.4 months (IQR, 10.5–44.0 months).

In Table 1, preoperative clinical data for patients undergoing RARP are displayed by the presence or absence of PLND. Although the follow-up period was significantly longer in the PLND group than in the non-PLND group, there were no significant differences in other covariates in either group.

### 3.2. Surgical and Pathological Outcomes

The surgical and pathological outcomes of this study are shown in Table 2. There were no statistically significant differences between the two groups.

Perioperative complications related to PLND were observed in four patients (0.7%), including grade 3 lymphorrhea in one patient (0.2%) and grade 3 lymphocele with infection in three (0.5%) [22].

### 3.3. Oncological Outcomes

At the end of follow-up period, there were no PCa deaths among the enrolled patients, but 12 (1.0%) died of other causes (details unknown). BCR after RARP was observed in 71 patients (5.9%), and the median time from RARP to BCR was 14.9 months. The 2- and 3-year BRFS rates were 95.0% and 92.4%, respectively.

The relationship according to NCCN risk stratification is shown in Figure 1. The 2-year BRFS rates were 98.8%, 94.9%, and 91.2% in the low-, intermediate-, and high-risk groups, respectively. The 3-year BRFS rates were 98.8%, 91.5%, and 88.5% in the low-, intermediate-, and high-risk groups, respectively. Patients with low-risk PCa had significantly higher BRFS than that of their counterparts (intermediate-risk PCa, *p* = 0.012; high-risk PCa, *p* = 0.001).

The relationship between BCR and patients with or without PLND is shown in Figure 2. The 2- and 3-year BRFS rates were 95.8% and 93.7%, respectively, in the non-PLND group and 94.3% and 91.5%, respectively, in the PLND group (*p* = 0.855).

Figure 3 shows the BRFS according to NCCN risk stratification. In the all-risk group, there were no significant differences between patients who underwent PLND and those who did not.

## 4. Discussion

Overall, the results of this study using propensity score-matched analysis suggest that omitting PLND when performing RARP for PCa may not affect oncological outcomes, especially BRFS. However, this was a retrospective multicenter cohort study, so the results should be interpreted with caution because of the significant differences in the surgical procedure, patient background, or extent of dissection depending on time and situation. In comparison with before propensity score matching, the PLND group accounted for 67.7% (1724 patients) of enrolled patients. In addition, the PLND group had significantly higher PSA and biopsy GG than the non-PLND group and were more often classified as high-risk PCa (*p* < 0.001). The median number of lymph nodes removed in patients who underwent ePLND was 16 (IQR, 11–21). Therefore, it seems unlikely that the surgical quality of PLND may be low. On the other hand, the BCR rate was significantly lower in the non-PLND group compared to the PLND group (*p* < 0.001). The non-PLND group had a significantly lower BCR rate than the PLND group for high-risk PCa (*p* = 0.004), although there was no significant difference between the two groups for intermediate-risk PCa (*p* = 0.230). Additionally, patients who underwent limited PLND had significantly lower BCR rates than those who underwent ePLND (*p* < 0.001). Due to the potential for greater bias due to variations in patient background, this study used propensity score matching to align patient backgrounds and compare the significance of PLND.

Whether PLND should be performed during RP for PCa is controversial. The benefit of PLND, especially ePLND, would undoubtedly contribute to improving the accuracy of PCa staging [5]. Additionally, this can help identify patients who require or omit further adjuvant therapy and predict postoperative oncological outcomes such as BCR or metastasis [5,6]. Although EAU and NCCN guidelines currently recommend ePLND for intermediate- or high-risk PCa patients, the therapeutic significance of PLND still needs to be discussed [5]. When selecting patients to undergo PLND using several nomograms [7,23], it is important for clinicians to consider the risk profile of the model development subject as well as the range of PLNDs to be utilized to establish the predictive ability of the nomograms [2,3]. Furthermore, several guidelines recommend various thresholds of nomogram predicted LVI probability for PLND at the time of RP [3]. Therefore, the calculated risk of patients harboring positive lymph nodes should be discussed, along with the utility of establishing the presence of LVI to inform future management and risks associated with PLND, and to share decision-making methods when performing PLND [3].

Although the evidence of benefits in PCa has been inconsistent and has suffered from methodological biases, with PLND often being performed at the time of RP, it is important to recognize that more extensive nodal dissection may be required for patients with an increased risk of LVI [24]. In a multi-institutional study of 94 patients with locally advanced PCa, ePLND without immediate androgen deprivation therapy was associated with favorable BRFS and metastasis-free survival (MFS) at the 2-year follow-up [25]. Abdollah et al. further reported a 10-year cancer-specific survival (CSS) of 74.7% to 97.9% for patients who underwent removal of 8–45 LNs, suggesting that removing more LNs was associated with improved CSS [26]. Bivalacqua et al. found that ePLND was significantly more advantageous than limited PLND in terms of 5-year BRFS and 10-year MFS [27]. A recent retrospective study enrolling 27,690 patients using the Surveillance, Epidemiology, and End Results database found that, in a multivariate analysis, an increase in the number of resected LNs was associated with an improvement in OS but not CSS [14]. A recent study identified the following lymphatic drainage pathways from the prostate: (A) the lymph vessels followed the medial umbilical ligament, continued laterally to the junction between the internal and external iliac arteries, and proceeded cranially; (B) the lymph vessels originating from the deep dorsal to the prostate followed the inferior vesical artery, and then proceeded laterally; and (C) the lymph vessels ran medial to the internal iliac and then to the common iliac arteries, thereafter proceeding cranially [28]. Since lymphatic flow from the prostate varies from patient to patient, ePLND may be useful in terms of cancer control in PCa.

In contrast, PLND was not meaningful in patients who had already undergone LNI, even though ePLND was a useful procedure in patients without LNI [6]. In a prospective randomized trial from the Memorial Sloan Kettering Cancer Center, 1440 patients with RP who underwent ePLND or limited PLND were included in the final analysis [9]. The median number of sampled LNs was 12 for limited PLND and 14 for ePLND, with corresponding LNIs of 12% and 14%, respectively [9]. At a median follow-up of 3.1 years, there were no significant differences in BCR rates between the two groups [9]. Lestingi et al. conducted a prospective single-center phase III trial in patients with intermediate- or high-risk clinically localized PCa [10]. This study concluded that ePLND provided better pathological staging and showed no significant differences in early oncological outcomes [10]. Moreover, in a subgroup analysis, patients with pathological GG 3–5 might show a potential benefit with improved BRFS [10]. A systematic review including 43 studies that enrolled 275,269 patients showed that PLND was associated with worse intra- and perioperative outcomes, whereas a direct therapeutic effect remained unclear [29]. In this study, no significant differences were observed in patients with PCa who underwent PLND after propensity score matching. Additionally, a relationship between PLND and PCa risk classification was not identified in this study. Although there were some problems, such as the relatively small number of lymph nodes removed, the results of this study suggest that PLND may not contribute to BCR in patients at any risk of PCa.

Another limitation is that PLND may have a high rate of surgery-related complications. Based on a comprehensive systematic review and meta-analysis, the perioperative morbidity of PLND in patients undergoing RP and PLND for PCa is highly dependent on the extent of PLND [11]. Fujimoto et al. previously reported that PLND during RP is recommended by clinical guidelines and is arguably the most accurate nodal staging method; however, PLND requires additional operative time and is associated with a greater risk of complications, such as lymphadenopathy [13]. Conversely, it is unclear whether ePLND is more complicated than limited PLND, even though PLND has a higher incidence of complications, particularly lymphorrhea or lymphocele formation [30]. The incidence of PLND-related complications was very low, and there was no significant increase in operative time in the PLND group. The risk of complications from PLND can be avoided by technical improvements based on the results of this study.

This study has several limitations. First, the retrospective design of this multicenter cohort study may have introduced a bias. Second, the pathology review, including biopsy and pathology GG, was not centralized in this study. Third, the relatively short follow-up period may have been insufficient to accurately identify the predictors of BCR after RARP. Finally, the number of dissected lymph nodes and the quality of PLND may not have been consistent, because the extent of lymph node dissection varied according to the surgeon’s preference or the policy of each institution.

## 5. Conclusions

To the best of our knowledge, this is the first large-scale retrospective multicenter cohort study in Japan investigating whether PLND may improve BRFS in patients with RARP. Based on the results of this log-rank study, we suggest that PLND may be unnecessary for patients with PCa undergoing RARP. Further prospective randomized trials and long-term evaluations are required to identify predictive factors for BCR in patients receiving RARP.

## Figures and Tables

**Figure 1 cancers-14-05803-f001:**
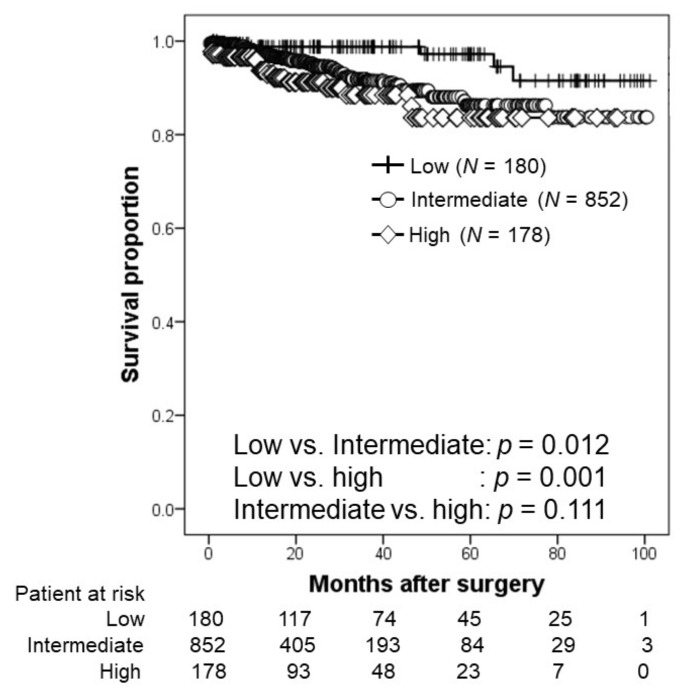
Kaplan–Meier estimates of biochemical recurrence-free survival according to the National Comprehensive Cancer Network risk stratification. The 2-year BRFS rates in the low-, intermediate-, and high-risk groups were 98.8%, 94.9%, and 91.2%, respectively.

**Figure 2 cancers-14-05803-f002:**
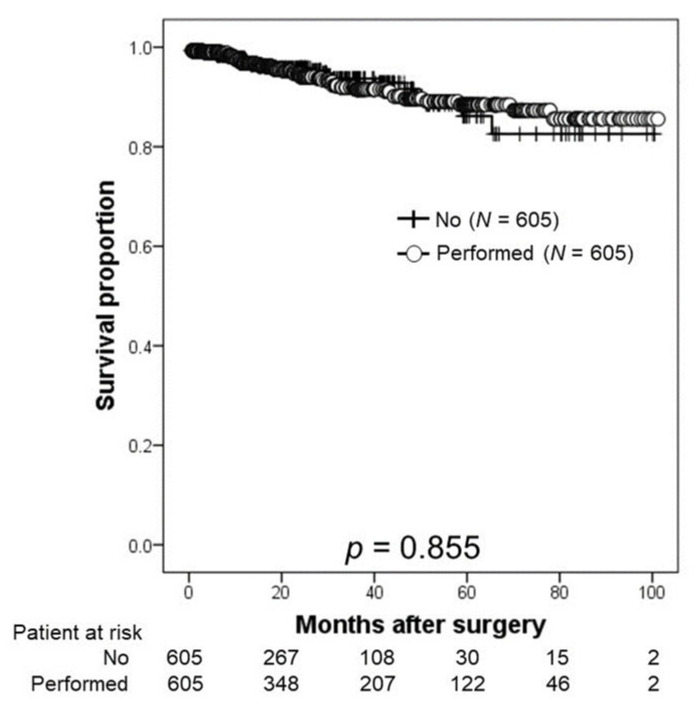
Kaplan–Meier estimates of biochemical recurrence-free survival according to the presence or absence of pelvic lymph node dissection (PLND). The 2-year BRFS rate was 95.8% in the non-PLND group and 94.3% in the PLND group.

**Figure 3 cancers-14-05803-f003:**
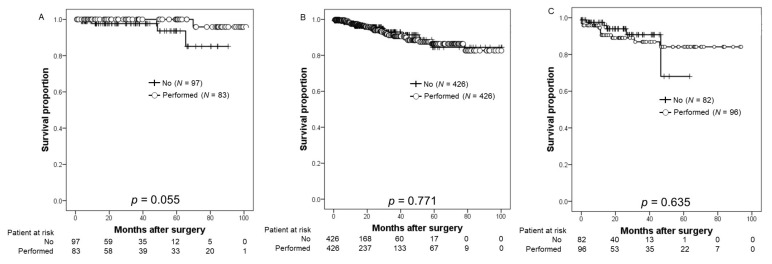
Kaplan–Meier estimates of biochemical recurrence-free survival (BRFS) according to the initial PSA levels stratified by the National Comprehensive Cancer Network risk stratification. (**A**) According to the low-risk PCa, the 2-year BRFS rate was 97.7% in patients undergoing robot-assisted radical prostatectomy (RARP) and pelvic lymph node dissection (PLND) (the non-PLND group) and 100% in those who underwent RARP with pelvic lymph node dissection (PLND) (PLND group). (**B**) Regarding the intermediate-risk PCa, the 2-year BRFS rates in the non-PLND and PLND groups were 95.7% and 94.3%, respectively. (**C**) Based on the high-risk PCa, the 2-year BRFS rate was 94.0% in the non-PLND group and 83% in the PLND group.

**Table 1 cancers-14-05803-t001:** Patient demographics.

Covariates	Non-PLND Group	PLND Group	*p*
**Number**	605	605	
**Age (median, year, IQR)**	69 (64–73)	68 (65–72)	0.967
**Body mass index (median, kg/m^2^, IQR)**	23.4 (21.7–25.2)	23.5 (21.7–25.6)	0.200
**Initial PSA (median, ng/mL, IQR)**	7.002 (5.280–10.220)	7.230 (5.300–10.344)	0.454
**Biopsy Grade group (number, %)**			0.541
**1**	170 (28.1)	168 (27.8)	
**2**	272 (45.0)	265 (43.7)	
**3**	115 (19.0)	121 (20.0)	
**4**	40 (6.6)	39 (6.4)	
**5**	8 (1.3)	13 (2.1)	
**Clinical T stage (number, %)**			0.751
**T1**	132 (21.8)	135 (22.3)	
**T2**	463 (76.5)	453 (74.7)	
**T3**	10 (1.7)	18 (3.0)	
**NCCN risk classification (number, %)**			0.139
**Low**	97 (16.0)	84 (13.9)	
**Intermediate**	426 (70.4)	426 (70.2)	
**High**	82 (13.6)	96 (15.9)	
**ECOG-PS (number, %)**			0.244
**0**	591 (97.7)	586 (96.7)	
**1**	14 (2.3)	19 (3.1)	
**2**	0	1 (0.2)	
**Follow-up period (median, months, IQR)**	17.2 (8.9–32.6)	24.7 (11.4–53.6)	<0.001

PLND, pelvic lymph node dissection; IQR, interquartile range; PSA, prostate-specific antigen; NCCN, National Comprehensive Cancer Network; ECOG-PS, Eastern Cooperative Oncology Group; T stage, tumor stage.

**Table 2 cancers-14-05803-t002:** Surgical and pathological outcomes.

Covariates	Non-PLND Group	PLND Group	*p*
**Number**	605	605	
**Console time (median, minutes, IQR)**	153 (116–213)	157 (120–201)	0.218
**Estimates blood loss (median, mL, IQR)**	50 (10–200)	50 (10–168)	0.517
**Pathological Grade group (number, %)**			0.756
**1**	55 (9.1)	62 (10.2)	
**2**	309 (51.0)	306 (50.5)	
**3**	167 (27.6)	167 (27.6)	
**4**	50 (8.3)	43 (7.1)	
**5**	24 (4.0)	28 (4.6)	
**Pathological T stage (number, %)**			0.763
**T2**	133 (22.0)	125 (20.7)	
**T3**	472 (78.0)	481 (79.3)	
**LNI (number, %)**	Not applicable	6 (1.0)	
**Lymph node count (number, median, IQR)**	Not applicable	5 (3–9)	
**Positive surgical margin (number, %)**	125 (20.7)	122 (19.8)	0.831

PLND, pelvic lymph node dissection; IQR, interquartile range; LNI, lymph node involvement; T stage, tumor stage.

## Data Availability

The data presented in this study are available on request from the corresponding author. The data are not publicly available due to privacy and ethical reasons.

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
