# Peer review of "Pelvic Lymphadenectomy May Not Improve Biochemical Recurrence-Free Survival in Patients with Prostate Cancer Treated with Robot-Assisted Radical Prostatectomy in Japan (The MSUG94 Group)"

_cancers, 2022, doi:10.3390/cancers14235803_

Round 1

Reviewer 1 Report

Overall, the paper is well put together, with a thorough research protocol.

The introduction presents the current guidelines regarding PLND in PCa, as well as the controversial evidence in terms of progression free and overall survival. For this section, I would recommend the mentioning of the percentage of positive lymph nodes in patients with cN0 but high risk of lymph node involvement on nomograms, as well as an approximation of the extra OR time needed to perform PLDN.      

            For Materials and methods section, I would suggest that the authors detail the inclusion and exclusion criteria for the selected patients, as the authors report that 3195 cases were analyzed retrospectively, but only 1210 patients were further enrolled.

            The Results section elaborate valuable findings, supported by thorough statistical analysis. However, the number of patients stratified as low-, intermediate- and high-risk does not correspond with the total of 1210 enrolled cases (rows 147 – 152).   

Author Response

November 19, 2022

Dear Editor:

Thank you very much for the review of our manuscript titled “Pelvic Lymphadenectomy may not Improve Biochemical Recurrence-Free Survival in Patients with Prostate Cancer Treated with Robot-Assisted Radical Prostatectomy in Japan (The MSUG94 Group).”

We sincerely appreciate all valuable comments and suggestions, which helped us to improve the quality of our manuscript. Our responses to the Reviewers’ comments are described below in a point-to-point manner. Appropriate changes, suggested by the Reviewers, have been introduced to the manuscript (track-changes mode in the red color font). Let me emphasize our full readiness to make any further improvements to the manuscript.

We hope that our manuscript will be acceptable for publication in the Cancers.

We look forward to hearing from you.

Yours sincerely,

Takuya Koie

Department of Urology

Gifu University Graduate School of Medicine

1-1 Yanagido, Gifu, Gifu 501-1194, Japan

TEL.: +81-582-30-6338

FAX: +81-582-30-6341

e-mail: goodwin@gifu-u.ac.jp

Responses to the reviewer's comments

We would like to thank the Reviewers for taking the time and effort necessary to review the manuscript. We sincerely appreciate all the valuable comments and suggestions, which helped us to improve the quality of the manuscript.

Response to Reviewer 1

The authors appreciate the reviewer’s comments. The authors’ point-by-point responses to the comments are given below.

  1. Overall, the paper is well put together, with a thorough research protocol.

The introduction presents the current guidelines regarding PLND in PCa, as well as the controversial evidence in terms of progression free and overall survival. For this section, I would recommend the mentioning of the percentage of positive lymph nodes in patients with cN0 but high risk of lymph node involvement on nomograms, as well as an approximation of the extra OR time needed to perform PLDN. 

Response:

The authors have added the following sentence on line 71:

Using Briganti nomogram cutoffs of 5% and 7%, 195 (46.4%) and 235 (55.9%) patients could have avoided ePLND, while 4 (2.1%) and 6 (2.6%) could have missed LNI [8].

The authors have added the following sentence on line 81:

In fact, PLND is reported to account for 10% of the operative time, and it is clear that the operative time is longer with PLND than without [13]. Colicchia et al. reported that ePLND was found to increase the operative time required for PLND (47 versus 26 min) compared to limited or standard PLND [14].

  1. For Materials and methods section, I would suggest that the authors detail the inclusion and exclusion criteria for the selected patients, as the authors report that 3195 cases were analyzed retrospectively, but only 1210 patients were further enrolled.

Response:

The authors have revised the following part on line 138:

Comparing the PLND and non-PLND groups, the PLND group had significantly higher initial PSA, biopsy GG (p <0.0001), and significantly more patients with clinical T3 and high-risk PCa (p <0.0001). For these reasons, we considered it likely that differences in patient background between the two groups would be reflected in the results of this study. Therefore, In this study, we developed a propensity score analysis to reduce the effects of selection bias and potential confounding factors in this study.

  1. The Results section elaborate valuable findings, supported by thorough statistical analysis. However, the number of patients stratified as low-, intermediate- and high-risk does not correspond with the total of 1210 enrolled cases (rows 147 – 152).

Response:

The authors have revised the following part on line 159:

high-risk PCa risk were found in 181 1,811 (15.0%),

Reviewer 2 Report

Given the retrospective nature of the study quality how do you control for the extent/quality  of the lymph node dissection .  I would expect surgeon techique/experience would be highly variable and would impact the results.  LNI seem very low for 600 cases of ePLND.  Once again, questioning quality of LN dissection.  I also question criteria for non PLND VS PLND as there does not seem to be a difference in pathology grade or stage, which to me seems odd.  I would have expected differences if a "true" criteria was followed ie low grade low stage preponderance for the non PLND group and high grade, higher stage for PLND group.  One would also expect longer operative time for PLND group, which also brings into question the quality of lymph node dissection. 

Author Response

November 19, 2022

Dear Editor:

Thank you very much for the review of our manuscript titled “Pelvic Lymphadenectomy may not Improve Biochemical Recurrence-Free Survival in Patients with Prostate Cancer Treated with Robot-Assisted Radical Prostatectomy in Japan (The MSUG94 Group).”

We sincerely appreciate all valuable comments and suggestions, which helped us to improve the quality of our manuscript. Our responses to the Reviewers’ comments are described below in a point-to-point manner. Appropriate changes, suggested by the Reviewers, have been introduced to the manuscript (track-changes mode in the red color font). Let me emphasize our full readiness to make any further improvements to the manuscript.

We hope that our manuscript will be acceptable for publication in the Cancers.

We look forward to hearing from you.

Yours sincerely,

Takuya Koie

Department of Urology

Gifu University Graduate School of Medicine

1-1 Yanagido, Gifu, Gifu 501-1194, Japan

TEL.: +81-582-30-6338

FAX: +81-582-30-6341

e-mail: goodwin@gifu-u.ac.jp

Responses to the reviewer's comments

We would like to thank the Reviewers for taking the time and effort necessary to review the manuscript. We sincerely appreciate all the valuable comments and suggestions, which helped us to improve the quality of the manuscript.

Response to Reviewer 2

The authors appreciate the reviewer’s comments. The authors’ point-by-point responses to the comments are given below.

  1. Given the retrospective nature of the study quality how do you control for the extent/quality of the lymph node dissection. I would expect surgeon techique/experience would be highly variable and would impact the results. LNI seem very low for 600 cases of ePLND. Once again, questioning quality of LN dissection. I also question criteria for non PLND VS PLND as there does not seem to be a difference in pathology grade or stage, which to me seems odd. I would have expected differences if a "true" criteria was followed ie low grade low stage preponderance for the non PLND group and high grade, higher stage for PLND group. One would also expect longer operative time for PLND group, which also brings into question the quality of lymph node dissection.

Response:

We have added the following sentences on line 220:

In comparison before propensity score matching, the PLND group accounted for 67.7% (1724 patients) of enrolled patients. In addition, the PLND group had significantly higher PSA and biopsy GG than the non-PLND group and were more often classified as high-risk PCa (p < 0.001). The median number of lymph nodes removed in patients who underwent ePLND was 16 (IQR, 11-21). Therefore, it seems unlikely that the surgical quality of PLND may be low. On the other hand, the BCR rate was significantly lower in the non-PLND group compared to the PLND group (p < 0.001). The non-PLND group had a significantly lower BCR rate than the PLND group for high-risk PCa (p = 0.004), although there was no significant difference between the two groups for intermediate-risk PCa (p = 0.230). Additionally, patients who underwent limited PLND had significantly lower BCR rates than those who underwent ePLND (p < 0.001). Because of the potential for greater bias due to variations in patient background, this study used propensity score matching to align patient backgrounds and compare the significance of PLND.

Round 2

Reviewer 2 Report

Thank for your reply. I accept your explanation and revisions.